# The roles of nuclear orphan receptor NR2F6 in anti-viral innate immunity

Chen Yang[1], Chen-Yu Wang[1], Qiao-Yun Long[1], Zhuo Cao[2], Ming-Liang Wei[1], Shan-Bo Tang[1], Xiang Lin[1], Zi-Qi Mu[1], Yong Xiao[2], Ming-Kai Chen[2]*, Min Wu[1]*, Lian-Yun Li[1]*

1 Frontier Science Center for Immunology and Metabolism, Hubei Key Laboratory of Cell Homeostasis, Hubei Key Laboratory of Developmentally Originated Disease, College of Life Sciences, Renmin Hospital of Wuhan University, Wuhan University, Wuhan, China, 2 Department of Gastroenterology, Renmin Hospital of Wuhan University, Wuhan University, Wuhan, China

* kaimingchen@163.com (M-KC); wumin@whu.edu.cn (MW); lilianyun@whu.edu.cn (L-YL)

**Data Availability Statement:** The original deep sequencing data were submitted to SRA database, which can be accessed by BioProjectID: PRJNA819114 (https://www.ncbi.nlm.nih.gov/sra/?term=PRJNA819114); or by GEO accession

## Abstract

Proper transcription regulation by key transcription factors, such as IRF3, is critical for anti-viral defense. Dynamics of enhancer activity play important roles in many biological processes, and epigenomic analysis is used to determine the involved enhancers and transcription factors. To determine new transcription factors in anti-DNA-virus response, we have performed H3K27ac ChIP-Seq and identified three transcription factors, *NR2F6*, *MEF2D* and *MAFF*, in promoting HSV-1 replication. NR2F6 promotes HSV-1 replication and gene expression *in vitro* and *in vivo*, but not dependent on cGAS/STING pathway. NR2F6 binds to the promoter of *MAP3K5* and activates AP-1/c-Jun pathway, which is critical for DNA virus replication. On the other hand, *NR2F6* is transcriptionally repressed by c-Jun and forms a negative feedback loop. Meanwhile, cGAS/STING innate immunity signaling represses *NR2F6* through STAT3. Taken together, we have identified new transcription factors and revealed the underlying mechanisms involved in the network between DNA viruses and host cells.

## Author summary

DNA virus infection, such as KSHV and HSV-1, affects a large population of human beings and difficult to be cleared from human bodies. The change of enhancer activity on chromatin reflects the cellular response upon viral infections, and Epigenomic profiling of active enhancers has been found to be a powerful tool to identify key transcription factors in biological processes. Here, we have performed H3K27ac ChIP-Seq to determine new transcription factors in anti-DNA-virus response, and identified three transcription factors, *NR2F6*, *MEF2D* and *MAFF*, in promoting HSV-1 replication. Further studies show that NR2F6 promotes HSV-1 replication and gene expression *in vitro* and *in vivo*, but not dependent on cGAS/STING pathway. NR2F6 is critical for AP-1/c-Jun pathway activation through direct targeting the promoter of *MAP3K5*, which is critical for DNA virus replication. On the other hand, host cell controls the expression of *NR2F6 via* c-Jun, which forms a negative feedback loop in anti-DNA-virus response. Meanwhile, cGAS/STING

GSE249665 (https://www.ncbi.nlm.nih.gov/geo/query/acc.cgi?acc=GSE249665). All the other data are included within the article or the supplementary information.

**Funding:** This work was supported by National Key Research and Development Program of China (2023YFA0913400 to WM), National Natural Science Foundation of China (3217050383 to LLY; 81972647 to WM), and the Fundamental Research Funds for the Central Universities (2042022dx0003 to WM). The funders had no role in study design, data collection and analysis, decision to publish, or preparation of the manuscript.

**Competing interests:** The authors have declared that no competing interests exist.

innate immunity signaling pathway represses *NR2F6* expression through STAT3. Taken together, we have identified new transcription factors and revealed the underlying mechanisms involved in the interaction between DNA viruses and host cells.

## Introduction

Virus infection is one of the major threats to human health all over the world. In comparison with RNA viruses, infection of some DNA viruses, such as KSHV and HSV-1, are detected in a large population of human beings and difficult to be cleared due to its latent infection [1–3]. Innate immunity mediated by cGAS (cyclic GMP-AMP synthase) pathway is one of the key pathways in host cells fighting against DNA viruses [4–7]. The viral DNA sensor cGAS recognizes viral DNA and synthesizes the second messenger cGAMP to activate the downstream cascade. cGAMP activates the adaptor protein MITA (also known as STING), which then recruits and promotes the phosphorylation of TANK binding kinase 1 (TBK1) and interferon regulatory factor 3 (IRF3) [8,9]. Phosphorylated IRF3 enters the nucleus and turns on type I interferon expression, which activates JAK/STAT pathway to express various anti-viral genes for virus elimination [5,10]. DNA virus infections, such as HSV-1, usually lead to the change of thousands of host genes [11]. Besides cGAS-IFN signaling, several pathways are involved in the process. For example, AP-1/c-Jun pathway is reported to be critical in viral gene expression and replication [12]. It is possible other signaling pathways may be involved in anti-viral processes. Thus, it is important to further investigate new anti-viral mechanisms in human cells, which will be useful to develop new anti-viral strategy.

Epigenetic marks on chromatin are signatures for cell identity, which usually co-operate with transcription factors to regulate chromatin structure and transcription [13–15]. During viral infection, epigenetic regulation is critical for proper anti-viral gene expression and cell response. Multiple epigenetic enzymes have been shown involved in anti-viral processes. For example, histone H3K36me3 methylase SETD2 has been shown to be critical in the response to HBV [16]. H3K4 demethylase LSD1 regulates K63-linked polyubiquitination of RIG-I and the subsequent anti-viral response and IFNB1 expression [17]. Besides the classic innate immunity pathway, epigenetics is also involved in other processes related to viral infection. Inhibition of LSD1 results in heterochromatic suppression of the HSV-1 genome and subsequently affects viral infection [18,19]. SETD8, one enzyme for histone H4K20me, has been shown to methylate PCNA and regulate DNA virus replication [20]. Inhibition of H3K27me3 demethylase JMJD3 and UTX by a small molecule, GSK-J4, inhibits HSV-1 reactivation from sensory neurons [21]. Thus, it is important to further explore the epigenetic mechanisms in host cell-virus interaction.

Recent epigenomic progresses have shown that changes of enhancer activities are involved in multiple processes [22,23]. When a signaling pathway is activated, the active forms of transcription factors bind to their corresponding enhancers and turn on transcription. The active enhancers are marked with histone modifications, such as H3K4me1 and H3K27ac, and have recently been widely studied in many disease models, especially in cancer research [14,15,24–30]. Nowadays, H3K27ac ChIP-Seq has been widely used to identify active enhancers in cells and tissues [13,29,31,32]. Since H3K27ac is distributed both on promoters and active enhancers, usually only the H3K27ac peaks in the distant regions to transcription start sites are used in enhancer analysis (ref??). The variant active enhancers between experimental groups are used to predict potential bound transcription factors, which is critical in identifying new transcription factors and signaling pathways [29,33].

To identify the new transcription factors involved in anti-DNA virus response, we performed H3K27ac ChIP-Seq analysis in THP-1 cells infected with HSV-1. We identified NR2F6 as one of the important host factors involved in the signaling network activated by viral infection through interaction with JAK/STAT and AP-1/c-Jun pathways. Our work reveals new mechanisms in the interacting network of DNA viruses and host cells, and provides potential new targets for future drug design.

## Results

### Genome-wide enhancer analysis of HSV-1 infected cells

THP-1 is a monocyte cell line and often used in the studies of innate immune response, because cGAS/STING pathway responds normal to DNA viruses. To identify new factors involved in anti-DNA-virus response, we infected THP-1 cells with HSV-1 (MOI = 1) for 24 hr, and performed RNA-Seq and H3K27ac ChIP-Seq with the infected and control cells, followed by bio-informatic analysis (Fig 1A). HSV-1 infection changed the expression of many genes; and totally we have identified 1065 up-regulated different expressed genes (DEGs) and 1894 down-regulated DEGs (FC $\geq$ 2, p value $\leq$ 0.05) (Fig 1B). Functional analysis showed that the up-regulated DEGs were mostly enriched in anti-viral pathways, and the down-regulated DEGs in various processes, such as cell division, protein folding, and metabolisms (Fig 1C). We predicted the active enhancers in the two group cells based on the H3K27ac peaks faraway from TSS [29,34], then further compared the enhancers in two groups and predicted the potential involved transcription factors (TFs) based on the different enhancer regions (Fig 1D). The most enriched TFs belonged to the IRF family, which conformed to our expectation (Fig 1D). We then took the predicted TFs and overlapped them with the DEGs, and identified 23 TFs potentially involved in the anti-DNA-virus response (Fig 1E and S1 Table). Among them, we could find the well-known anti-viral TFs, such as IRF1/2/7 and STAT1/2, supporting the accuracy of our analysis (Fig 1E).

### Validation of transcription factors involved in anti-viral response

To validate the predicted TFs, we selected 5 TFs not well studied in the anti-viral research, designed siRNAs and successfully knocked down three of them, *Nr2f6*, *Mef2d* and *Maff*, in MEF cells. We then checked the expression of viral gene *US11* with qRT-PCR, and found that the knockdown of the three genes all down-regulated *US11* expression, suggesting they might be involved in anti-viral response (Figs 1F and S1). Interestingly, HSV-1 infection repressed *Nr2f6* mRNA in MEF cells (Fig 1F). We further exogenous expressed HA-tagged *Nr2f6* in MEF cells and found that its over-expression increased the expression of viral genes and the amount of viral DNA (Fig 1G).

NR2F6 is an orphan nuclear receptor involved in the regulation of T cells and anti-tumor immunity [35–38]. Since it is known as a regulatory gene of immune system, we are interested to study its roles and the underlying mechanisms in anti-tumor response. To investigate the function of *NR2F6* in multiple cell lines, we knocked it down in THP-1, U2OS and HFF cell lines, and found that *NR2F6* deficiency caused down-regulation of HSV-1 *US11* gene in all the tested cell lines (Figs 2A, S2A and S2B). On the contrary, exogenous expression of *NR2F6* in THP-1 increased HSV-1 replication (Fig 2B). We then investigated the function of NR2F6 when cells were infected by other viruses. We knocked down *NR2F6* in U2OS and HFF cells and infected with an RNA virus, SeV (Sendai virus). Quantitative RT-PCR results indicated that *NR2F6* knockdown repressed SeV in the above cell lines (S2A and S2B Fig). We then over expressed *NR2F6* in MEF and HEK293T cell lines, and *NR2F6* expression increased the amount of SeV in the cells (S2C and S2D Fig). We also knocked *Nr2f6* down in MEF cells, and

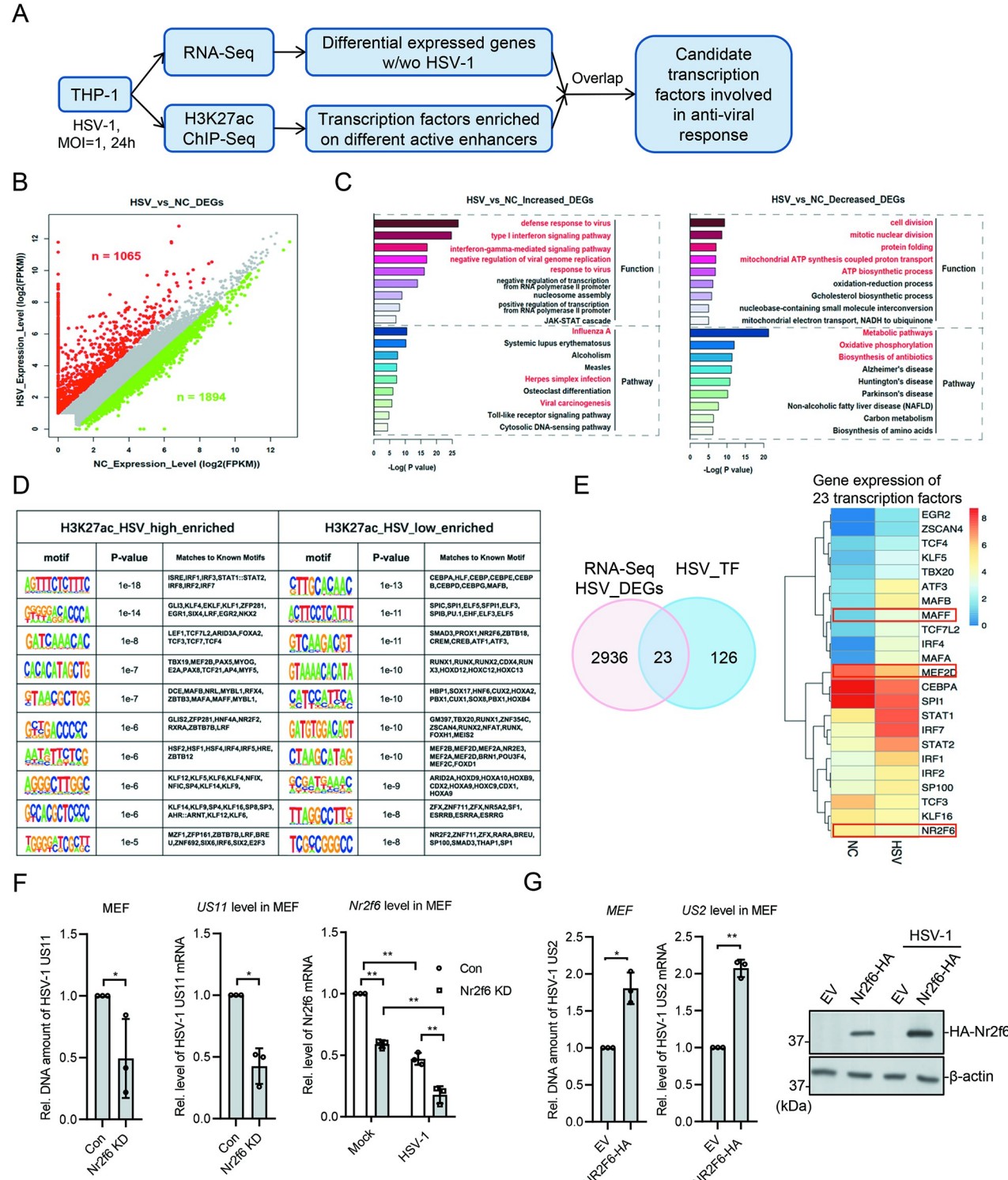

**Fig 1. Identification of anti-viral transcription factors.** (**A**) The pipeline for transcription factor identification. (**B**) RNA-seq analysis for DEGs in THP-1 after HSV-1 infection. Gene expression level was normalized by FPKM. A threshold of (two-fold change and −log10 p-value) was used for defining significant changes. Red for up-regulated genes, and green for down-regulated genes. (**C**) Biological process and KEGG analysis for the increased (left) and decreased DEGs (right). (**D**) The predicted top TFs enriched on the variant enhancer loci. (**E**) The Venn diagram showed DEGs overlapped with predicted TFs. The heatmap showed the relative expression level of the 23 overlapped TFs. (**F**) MEF cells were transfected with control or *Nr2f6* siRNA for 48 h. The MEF cells were infected with HSV-1 (MOI = 1) for 8 h, and the DNA amount of HSV-1 *US-11* gene, its mRNA

and the host *Nr2f6* mRNA level were measured by qPCR. **(G)** MEF cells were transfected with *Nr2f6* expression plasmids and then infected with HSV-1 for 8 h. The DNA amount of HSV-1 *US2* gene and the mRNA of viral *US2* gene were measured by qPCR. The expressed Nr2f6 level was measures by western blot. Graphs show mean ± SEM, n = 3. ** stands for p < 0.01, * for p < 0.05.

infected the cells with another DNA virus, MCMV (mouse cytomegalovirus). *Nr2f6* deficiency also successfully repressed MCMV replication and gene expression (S2E Fig). The above data indicated that NR2F6 promotes the replication and gene expression of multiple types of DNA viruses, as well as RNA virus SeV.

## Impairment of HSV-1 replication in *Nr2f6* deficient mice

To further investigate the *in vivo* function of Nr2f6 on viruses, we established *Nr2f6* knockout mice (S3A–S3C Fig). It was quite difficult for us to get enough homozygous mice, and we guessed the homozygous may encounter problems in breeding or development. Then we used heterozygous mice for our experiments. Compared with the wild type, $Nr2f6^{+/-}$ mice did not show a significant difference in liver weight, but the weight of spleen and lung was relatively lower (S3D–S3F Fig).

We isolated BMDMs from the bone marrows of *Nr2f6+/+* and $Nr2f6^{+/-}$ mice, and infected them with HSV-1. The gene expression and DNA amount of HSV-1 were impaired in the BMDMs from $Nr2f6^{+/-}$ mice (Fig 2C). We then intraperitoneal injected HSV-1 into the mice and isolated tissues to detect viral activity. We found that the expression of viral genes was decreased in the liver and lung tissues of $Nr2f6^{+/-}$ mice, while the expression of immune genes, such as *Ifnb1*, *Isg56*, *Cxcl10* and *Ifih1*, also decreased in the tissues from KO mice (Fig 2D–2F). The $Nr2f6^{+/-}$ mice also showed a better survival rate compared with the wild type upon the lethal dosage of HSV-1 virus (Fig 2G). All these indicate that Nr2f6 is involved in the positive regulation of HSV-1 replication and gene expression *in vivo* and *in vitro*.

## NR2F6 does not affect cGAS/STING anti-viral pathway

In mammalian cells, cGAS/STING pathway is considered as the most important pathway in anti-DNA-virus [39,40]. From the above results, we noticed that repression of virus replication by *Nr2f6* deficiency was not companied with increase of anti-viral genes (Fig 2F). To study whether NR2F6 regulates the expression of innate immune pathways, we performed RNA-Seq with *NR2F6* knockdown and wild type THP-1 cells. We analyzed the different expressed genes (DEGs) up-regulated upon HSV-1 infection, and found that among the DEGs up-regulated in *NR2F6*-deficient cells only very few were enriched in anti-viral pathways; while those down-regulated in *NR2F6*-deficient cells were enriched interferon-mediated anti-viral response (Fig 3A–3C and S2 Table). Since cGAS/STING pathway exerts anti-viral function mainly through interferon signaling, and plus the fact that cGAS/STING pathway is defective in U2OS cells [41] and *NR2F6* knockdown impaired HSV-1 replication in U2OS, we deduced that the repression of HSV-1 in *NR2F6*-deficient cells was not through cGAS/STING pathway; and the down-regulation of anti-viral genes was probably due to the lower level of HSV-1. We further checked it in U5A cells, one cell line with interferon signaling deficiency caused by *IFNAR2c* mutation, and found that *NR2F6* knockdown could still repress HSV-1 gene expression (Fig 3D). In THP-1 cells, either knocking down *NR2F6* with CRISPR/sgRNA, or establishing stable cells lines with *NR2F6* exogenous expression, we did not observe a significant difference of phosphorylated IRF3 and TBK1 in comparison with control cells (Fig 3E). In fact, the expression of *IFNB1*, *CXCL10* and *IL8* decreased in the *NR2F6* knockdown cells, but not the typical

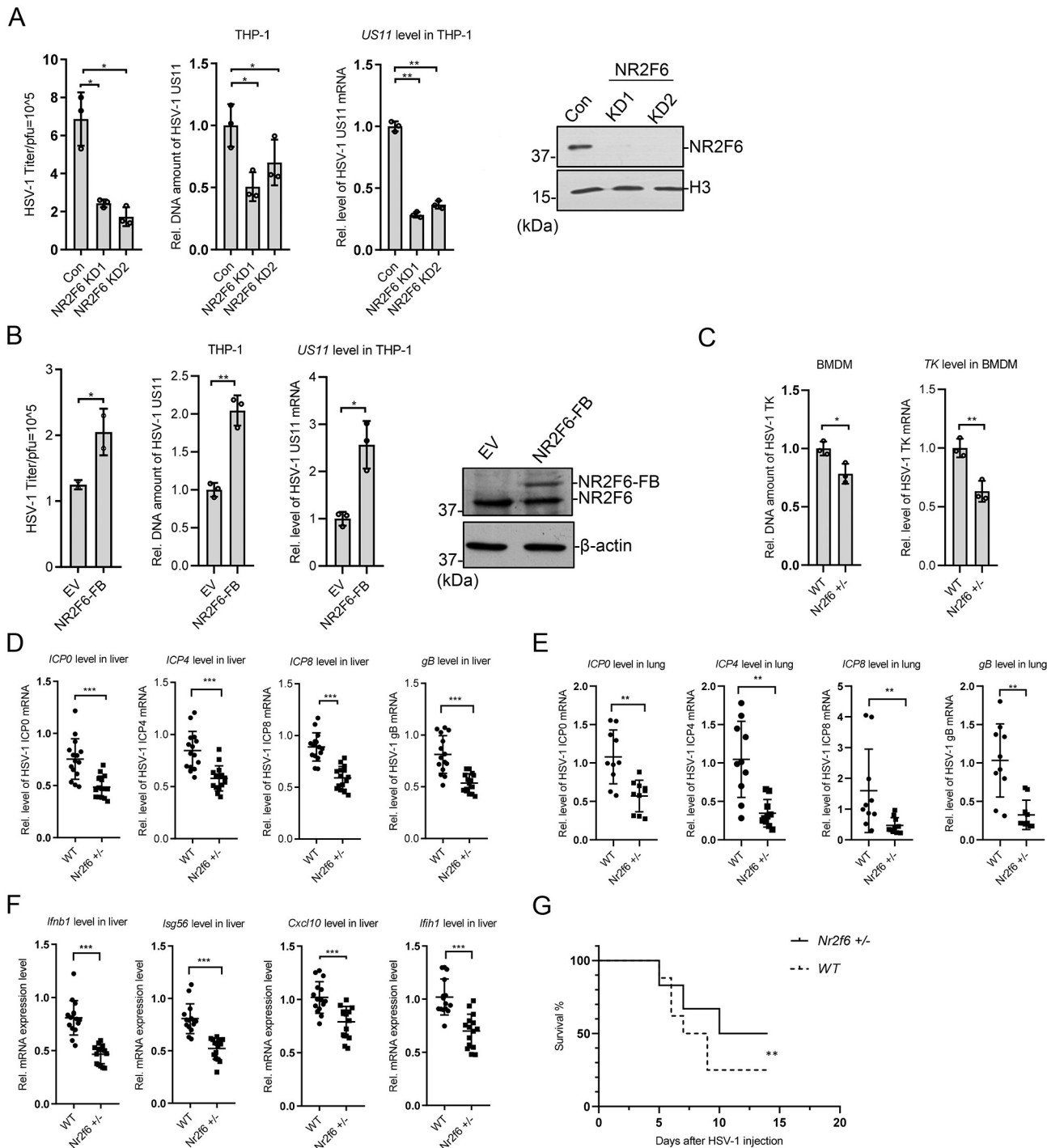

**Fig 2. NR2F6 positively regulates HSV-1 transcription and replication. (A)** *NR2F6* knockdown and control THP-1 cells were generated by CRISPR-Cas9 and selected with puromycin (2 μg/mL) for 48 h. The cells were infected with HSV-1 (MOI = 1) for 48 h before plaque assay or 24 h before qPCR (n = 3). Viral DNA and mRNA level were measured by qPCR, NR2F6 by immunoblotting. **(B)** *NR2F6* stable expressed THP-1 cells were screened with puromycin (2 μg/mL) for 48 h before immunoblotting. The cells were infected with HSV-1 (MOI = 1) for 48 h before plaque assay or 24 h before qPCR (n = 3). Viral DNA and mRNA level were measured by qPCR, NR2F6 by immunoblotting. **(C)** BMDM cells were isolated from the bone marrow of 8-week-old *Nr2f6*[+/+] and *Nr2f6*[+/-] mice, and infected with HSV-1 (MOI = 1) 8 h before qPCR analysis. Viral DNA and RNA were assayed by qPCR (n = 3). **(D-F)** Wild-type and *Nr2f6*[+/-] mice (n = 5) were infected with HSV-1 for 48 h by intraperitoneal injection. Expression of the viral genes in the liver (D) or lung (E) tissues, or anti-viral genes (*Ifnb1*, *Isg56*, *Cxcl10*, *Ifih1*) in liver (F), was assay by qPCR. **(G)** The survival of the above mice was monitored for 14 days. Graphs show mean ± SEM. ** stands for p < 0.01, * for p < 0.05.

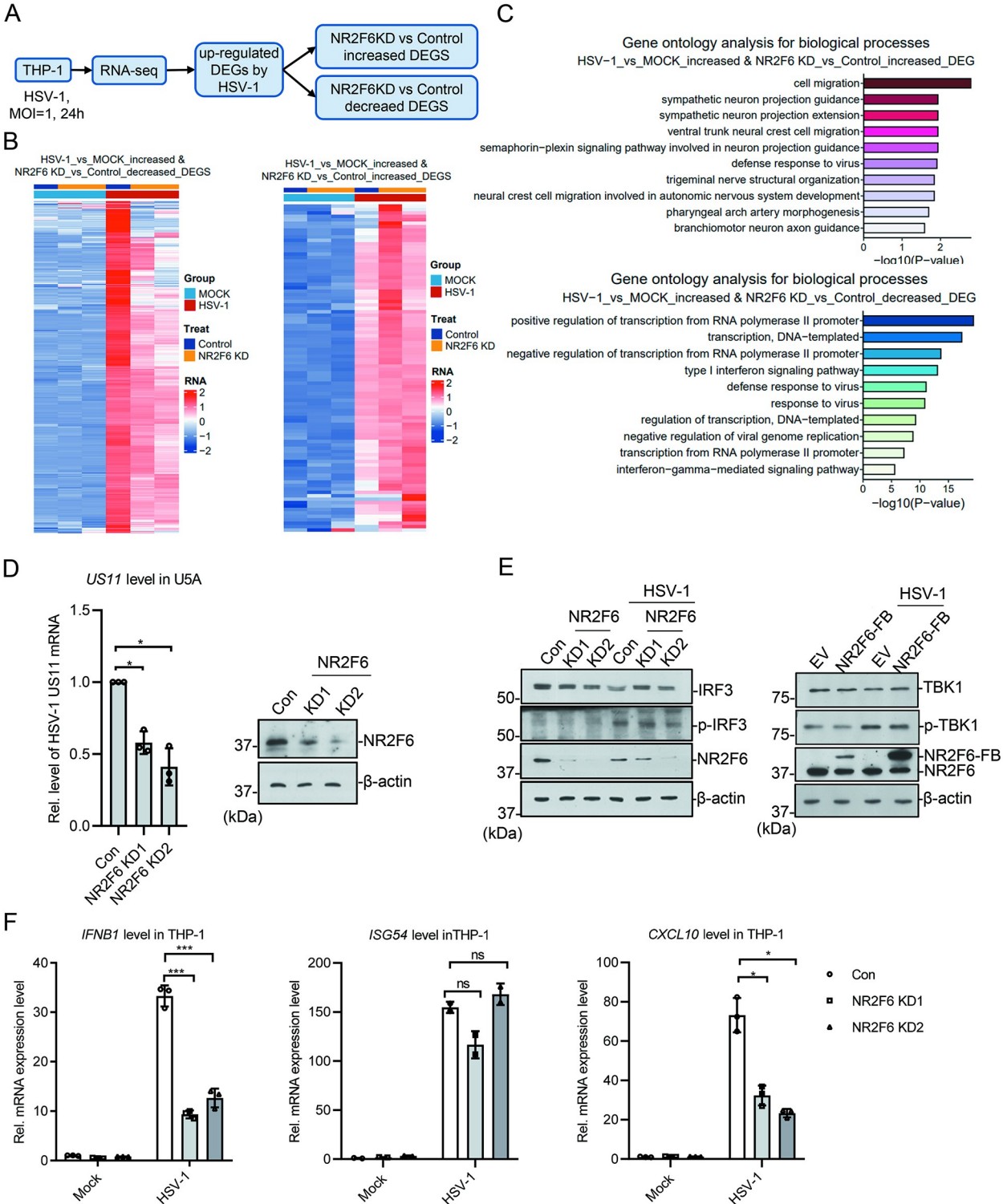

**Fig 3. The effect of NR2F6 to HSV-1 is not dependent on cGAS pathway. (A)** Pipeline for HSV-1-induced DEG analysis in NR2F6-knockdown THP-1. **(B)** The heat map of the HSV-1-induced up-regulated genes regulated by NR2F6. **(C)** Gene ontology analysis for the DEGs identified in (B). **(D)** *NR2F6* was knocked down in U5A cells by CRISPR. The cells were infected with HSV-1 (MOI = 1) for 8 h before qPCR analysis. NR2F6 level was measured by immunoblotting. **(E)** *NR2F6* was knocked down or expressed in THP-1 cells. The cells were infected with HSV-1 (MOI = 1) for 24 h and the indicated proteins were assayed with western blotting. **(F)** *NR2F6* knockdown cells were infected with HSV-1 (MOI = 1) for 24 h, and the mRNAs of *IFNB1*, *ISG54* and *CXCL10* were assayed with qPCR. Graphs show mean ± SEM, n = 3. ** stands for p < 0.01, * for p < 0.05.

interferon-inducible genes, such as *ISG54* and *ISG56* (Figs 3F and S4A). Taken together, these indicate that the regulation of HSV-1 by NR2F6 is not through cGAS/STING pathway.

## NR2F6 regulates HSV-1 replication through AP-1/c-Jun pathway

Among the down-regulated DEGs in *NR2F6* knockdown cells, we found that a portion of DEGs were enriched in JNK pathway (Figs 4A and S4A). JNK pathway has been shown to play positive roles in HSV-1 replication [12]. We then first examined the function of NR2F6 to *c-Jun* expression. The results indicated that HSV-1 infection activated *c-Jun* expression, and *NR2F6* knockdown caused the reduction of *c-Jun* mRNA and protein, as well as the phosphorylated forms of c-Jun and JNK (Fig 4B). *NR2F6* expression increased *c-Jun* expression in THP-1, but not *IFNB1* and its downstream genes, and upregulated the phosphorylated forms of c-Jun and JNK (Figs 4C, S4B and S4C). To verify the function of JNK/c-Jun pathway on HSV-1 replication, we knocked down *c-Jun* in THP-1, and found that *c-Jun* deficiency repressed HSV-1 gene expression, as well as *IFNB1* expression (Figs 4D and S5A). In MEF cells, stable expression of FB-tagged *c-Jun* increased HSV-1 viral titer (Fig 4E). Upon SeV infection, *c-Jun* knockdown caused a decrease of SeV amount (S5B Fig). The results were similar to those got from NR2F6 knockdown experiments. T5224 and SP600125, two inhibitors for JNK pathway, could both repress HSV-1 gene expression (Fig 4F and 4G). Moreover, SP600125 treatment in THP-1 successfully repressed HSV-1 gene expression and DNA replication enhanced by *NR2F6* expression (Fig 4H). The above data indicate that NR2F6 promotes HSV-1 replication probably through AP-1/c-Jun pathways.

## Cellular localization of NR2F6

Since NR2F6 is a transcription factor, it should be localized in the nucleus. To verify it, we first expressed HA-tagged *Nr2f6* in MEF cells, separated the cytoplasm and nucleus fractions and performed western blotting. The results showed that HA-Nr2f6 was localized in both fractions and we did not observe an obvious difference after HSV-1 treatment (Fig 5A). The immunostaining results also supported it (Fig 5B). We also fractionated THP-1 cells and examined the localization of endogenous NR2F6. The endogenous protein was also distributed in both cytoplasm and nucleus, and NR2F6 level decreased in both fractions after HSV-1 treatment (Fig 5C). It suggests the endogenous NR2F6 is probably transcriptionally repressed by HSV-1. Interestingly, we observed two NR2F6 bands could be recognized by the current antibody, and the nuclear NR2F6 seemed smaller than the cytoplasmic form (Fig 5C). It is possible that alternative splicing or post-translational modification might be involved in the regulation.

## NR2F6 regulates JNK pathway through targeting *MAP3K5*

To further explore the mechanisms of NR2F6 regulating JNK pathway, we used the online NR2F6 ChIP-Seq datasets, overlapped the potential NR2F6 target genes with the DEGs down-regulated in *NR2F6* knockdown cells, and identified four genes associated with JNK pathway cascade, including *MAP3K5* (also known as *ASK1*), *UBC*, *TRIB1* and *DUSP10* (Fig 5D). MAP3K5 is one of the well-known kinases regulating JNK pathway. We verified that *NR2F6* knockdown decreased *MAP3K5* mRNA level, which was activated by HSV-1 infection; and *NR2F6* exogenous expression activated *MAP3K5* expression (Fig 5E). The online ChIP-Seq data showed that NR2F6 might bind *MAP3K5* on chromatin (Fig 5F). The result of ChIP-PCR showed that FB-tagged NR2F6 did not bind *MAP3K5* without virus infection, and was recruited to *MAP3K5* promoter region upon HSV-1 treatment (Fig 5G). Further studies found that NR2F6 also directly bound UBC and TRIB1 promoters and regulated their expression (S6 Fig). We also found that c-Jun might bind *Map3k5* based on the online data (Fig 5H).

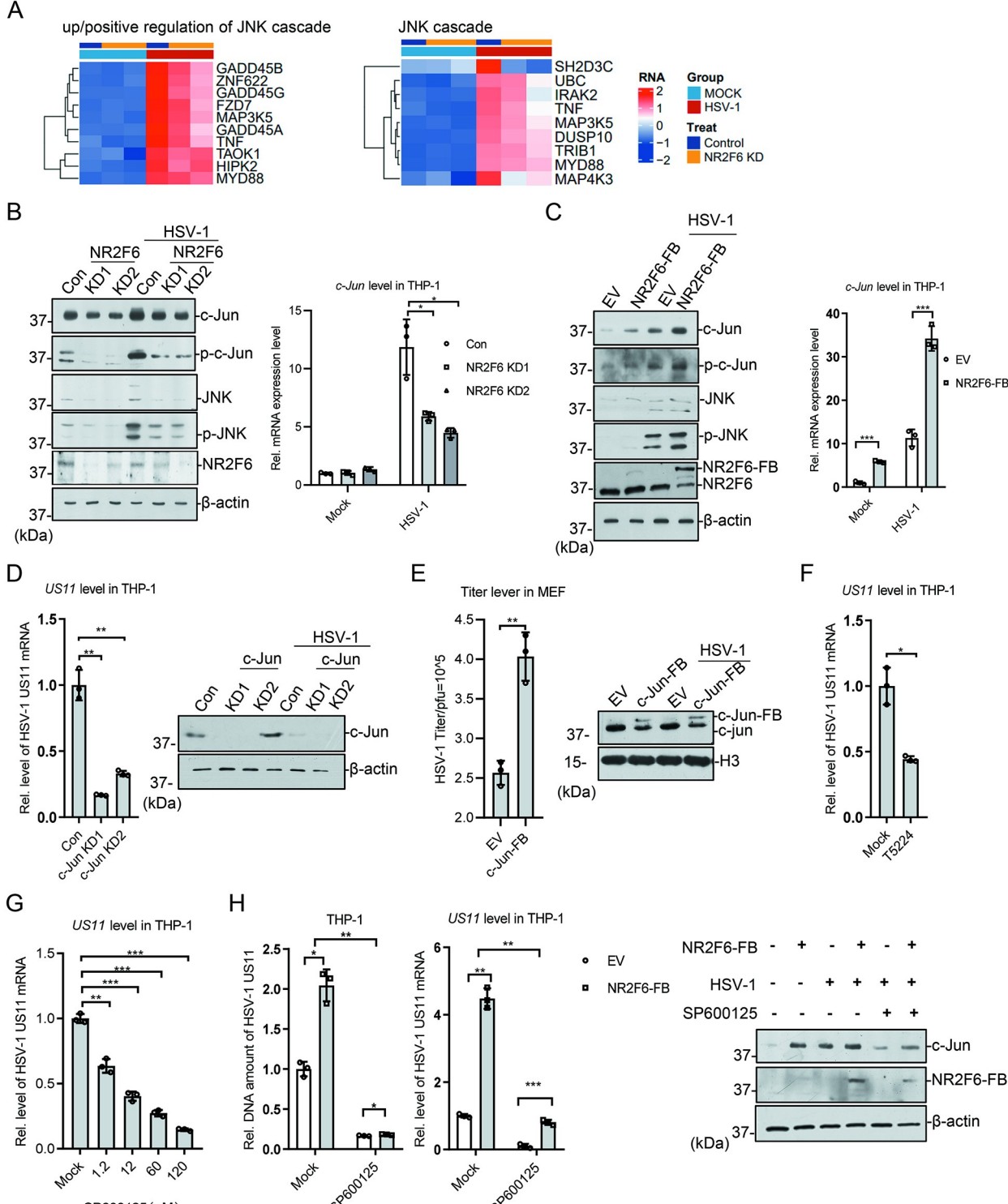

**Fig 4. NR2F6 promotes HSV-1 transcription and replication through JNK pathway. (A)** Heat map analysis showed that knockdown of NR2F6 down-regulated some genes in the JNK cascade after infection with HSV-1. **(B)** *NR2F6* knockdown cells were infected with HSV-1 (MOI = 1) for 24 h before immunoblotting of the indicated proteins and qPCR. **(C)** NR2F6 stable expressed THP-1 cells were infected with HSV-1 (MOI = 1) for 24 h before immunoblotting of the indicated proteins and qPCR. **(D)** *c-Jun* knockdown THP-1 cells were generated by CRISPR and the level of c-Jun was measured by immunoblotting. The cells were infected with HSV-1 (MOI = 1) for 24 h and *US11* viral gene expression was assayed by qPCR. **(E)** *c-Jun* was stably expressed in MEF cells. The cells were infected with HSV-1 (MOI = 1) for 24 h before plaque assay was performed. The

level of c-Jun was measured by immunoblotting. **(F)** THP-1 cells were treated with T-5224, the c-Fos inhibitor, or DMSO for 24h, and then infected with HSV-1 (MOI = 1) for 24 h before qPCR. **(G)** THP-1 cells were treated with different amount of SP600125, a JNK inhibitor, or DMSO for 24 h, and then infected with HSV-1 (MOI = 1) for 24 h before qPCR. **(H)** *NR2F6* stable expressed or control THP-1 cells were treated with SP600125 and infected with HSV-1 (MOI = 1) for 24 h before immunoblotting and qPCR. The viral DNA and RNA were assayed with qPCR. Graphs show mean ± SEM, n = 3. ** stands for $p < 0.01$, * for $p < 0.05$.

ChIP-PCR result showed that c-Jun was recruited to *Map3k5* promoter similar with NR2F6 (Fig 5I). These indicated that NR2F6 and c-Jun together regulate JNK pathway activation through *MAP3K5*.

## Repression of *NR2F6* expression by AP-1/JNK pathway

In the previous results, we found that HSV-1 treatment repressed NR2F6 expression (Figs 1E, 1F and 5C). To further confirm it, we examined the effect in several cell lines, including MLF, MEF, HFF, L929, THP-1, HACAT and HEK293T. The result showed that HSV-1 repressed *NR2F6* expression in all the above cell lines except HEK293T (Fig 6A). The immunoblots of HSV-1-infected THP-1 cells also indicated that HSV-1 repressed *NR2F6* expression, which was negative correlated with c-Jun phosphorylation (Fig 6B). Then we asked whether AP-1/ JNK signaling repressed *NR2F6* expression. We used c-Fos/AP-1 inhibitor T5224 to treat THP-1 cells and the qRT-PCR results showed that HSV-1 infection activated NR2F6 expression after T5224 treatment, instead of repressing it (Fig 6C). We also used the JNK inhibitor SP600125 to treat THP-1 cells (Fig 6D), and found that SP600125 treatment increased *NR2F6* expression in a dose-dependent manner. We then checked the online c-Jun ChIP-Seq data and found that c-Jun is possibly enriched on the *Nr2f6* promoter (Fig 6E). Then we verified it in a MEF cell line stably expressing FB-tagged c-Jun. The results of capture ChIP showed that c-Jun-FB was enriched on the *Nr2f6* promoter upon HSV-1 infection (Fig 6F). All these indicate that HSV-1 represses NR2F6 expression through AP-1/JNK signaling pathway.

## STAT3 represses *NR2F6* expression

Besides the above, we also tried to explore the potential relationship between innate immunity signaling with NR2F6 regulation. In an IRF3-FB stable cell lines, we found that exogenous IRF3 expression significantly repressed *NR2F6* mRNA and protein level (Fig 7A), suggesting HSV-1-mediated cGAS/STING signaling is also involved in the regulation of *NR2F6* expression. We then applied IFN-β to cells, and found that IFN-β treatment repressed *Nr2f6* expression (Fig 7B). We searched the online databases for the ChIP-Seq datasets of transcription factors involved in anti-viral innate immunity, such as IRF3, IRF7, NF-κB, STAT1/2/3. We found that STAT3 might be able to bind to *NR2F6* promoter (Fig 7C). We examined the status of STAT3 in our experimental system and found that the phosphorylated form of STAT3, which represented the active STAT3, increased after HSV-1 infection, while NR2F6 decreased (Fig 7D). The ChIP-PCR results using the phosphorylated STAT3 antibody showed that STAT3 binds to *NR2F6* promoter (Fig 7E). When *STAT3* was knocked down in the cells, NR2F6 mRNA level and HSV-1 gene expression increased (Fig 7F). The above results indicate that interferon signaling represses *NR2F6* expression to inhibit HSV-1 replication through STAT3.

## Discussion

Compared with RNA viruses, DNA viruses usually encode a relative larger number of genes, and change the transcription profile more dramatically in the host cells. Although many

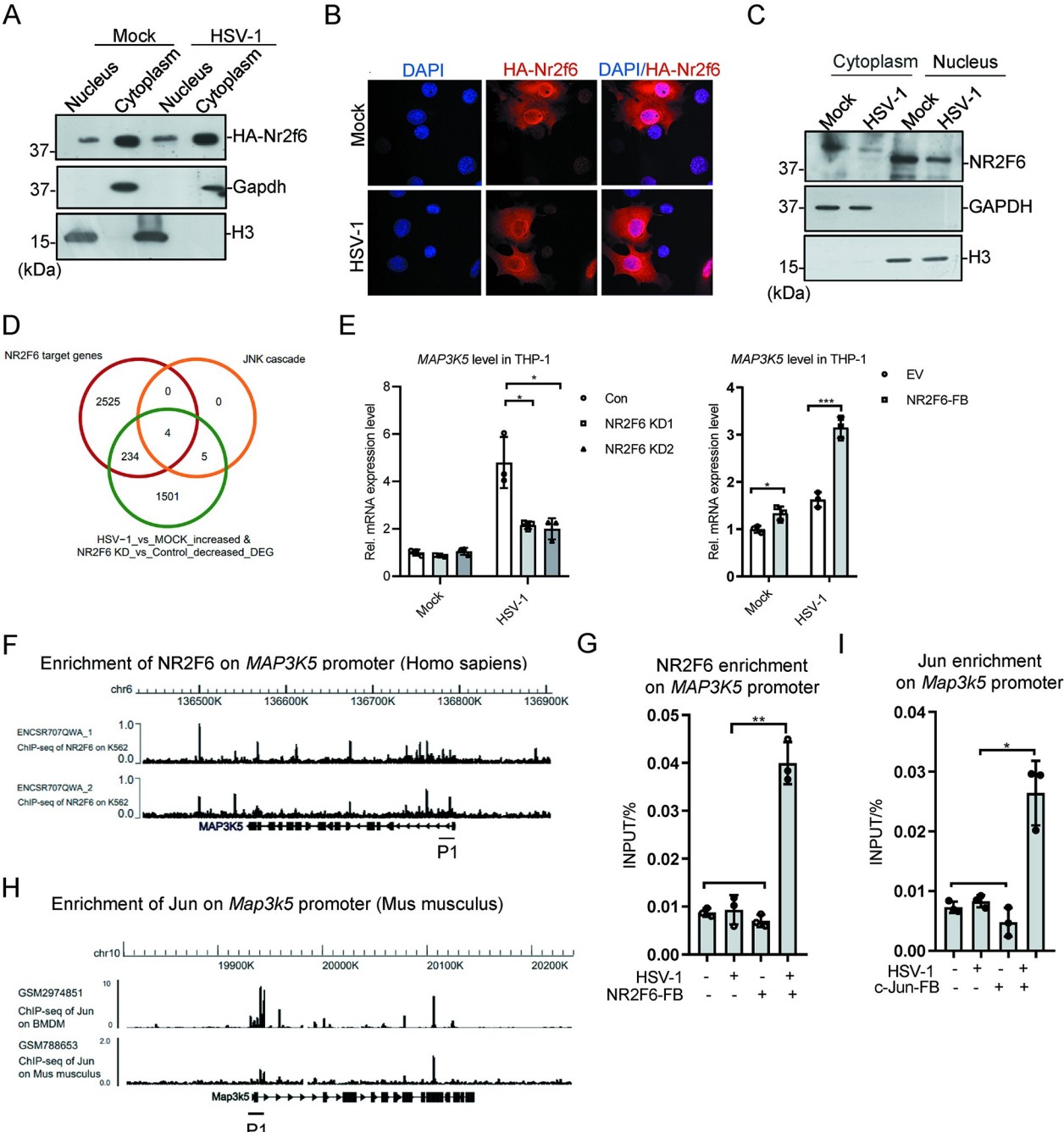

**Fig 5. NR2F6 enhances JNK signaling by directly targeting *MAP3K5*.** **(A)** HA-tagged *Nr2f6* was transiently expressed in MEF cells. The cells were infected with HSV-1 (MOI = 1) for 8 h, and the cytoplasm and nucleus fractions were separated, followed by western blotting of the indicated proteins. **(B)** *HA-Nr2f6* was transiently expression in MEF cells. The cells were infected with HSV-1 (MOI = 1) for 8 h before immune-staining. **(C)** THP-1 cells were infected with HSV-1 (MOI = 1) for 24 h. Then the cytoplasm and nucleus fractions were separated, followed by western blotting of the indicated proteins. **(D)** A Venn diagram to show the overlapping genes of the down-regulated genes in JNK pathway with *NR2F6* knocked-down, the down-regulated HSV-1-induced genes after *NR2F6* knockdown, and NR2F6-binding genes. **(E)** *NR2F6* knockdown (left) or stably expressed (right) THP-1 cells were infected with HSV-1 (MOI = 1) for 24 h, and *MAP3K5* mRNA was assayed with qPCR. **(F)** The UCSC genome browser view to show NR2F6 enrichment on *MAP3K5* promoter. **(G)** FB-tagged *NR2F6* was stably expressed in THP-1 and the cells were infected with HSV-1 (MOI = 1) for 24 h. Capture-ChIP was performed to examine NR2F6 enrichment on *MAP3K5* promoter. The primer target loci shown in (F). **(H)** The UCSC genome browser view to show that c-Jun enrichment on *Map3k5* promoter. **(I)** FB-tagged *c-Jun* was stably expressed in MEF and the cells were infected with HSV-1 (MOI = 1) for 8 h. Capture-ChIP was performed to examine c-Jun enrichment on *Map3k5* promoter. The primer target loci shown in (H). Graphs show mean ± SEM, n = 3. ** stands for p < 0.01, * for p < 0.05.

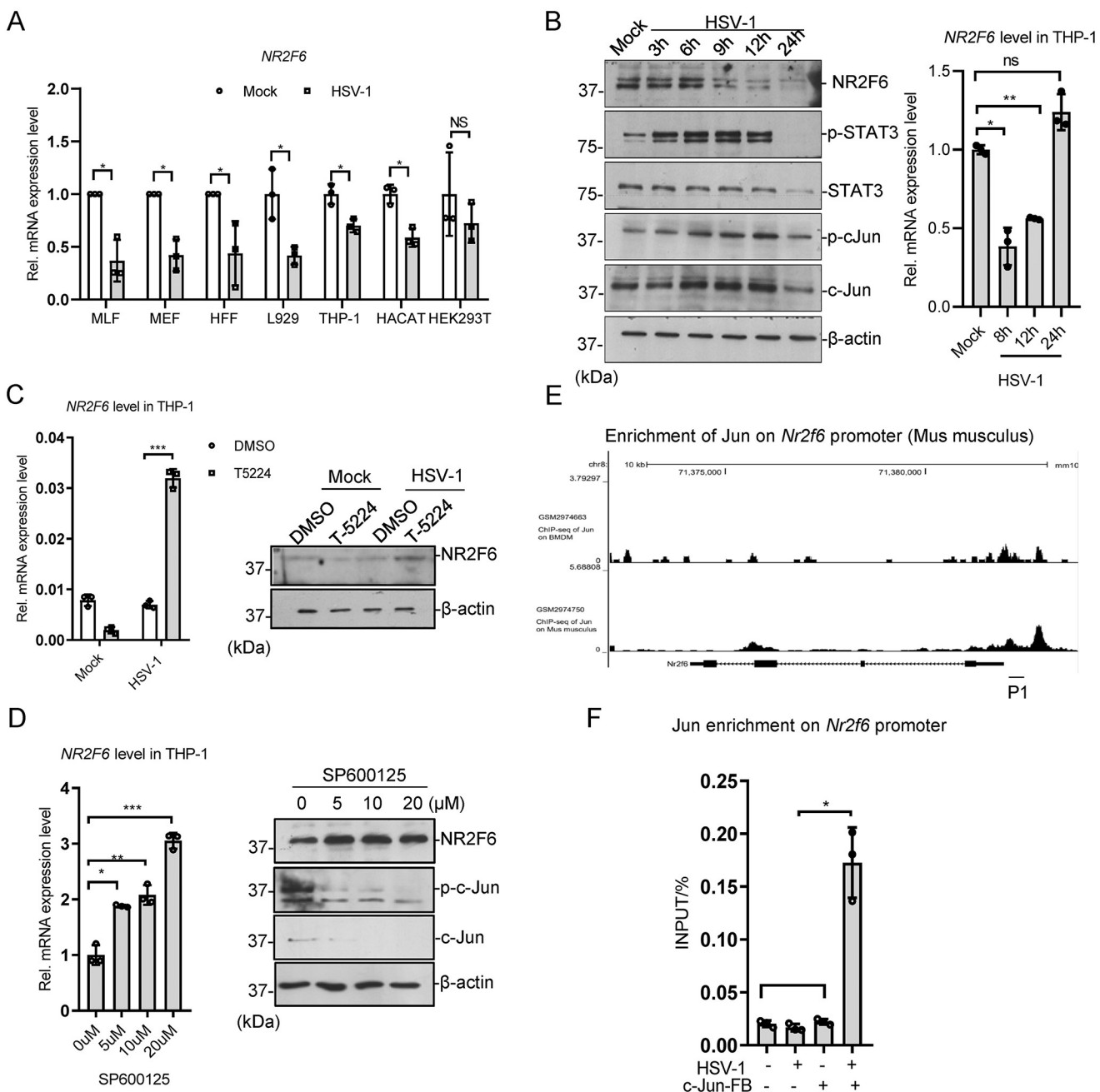

**Fig 6. AP-1/c-Jun pathway transcriptionally represses *NR2F6* upon HSV-1 infection.** (**A**) A variety of cells (MLF, MEF, HFF, THP-1, HACAT and HEK293T) were infected with HSV-1 (MOI = 1) for 8 h and NR2F6 mRNA were measured with qRT-PCR. (**B**) THP-1 cells were infected with HSV-1 (MOI = 1) for 24 h, and samples were collected at the indicated time points. Western blotting was performed with the indicated antibodies. *NR2F6* mRNA was measured with qRT-PCR. (**C**) THP-1 cells were treated with T-5224 or DMSO for 24 h, and then infected with HSV-1 (MOI = 1) for 24 h. Quantitative RT-PCR and western blotting were performed to measure NR2F6 level. (**D**) THP-1 cells were treated with the indicated amounts of SP600125 or DMSO for 24 h, and then infected with HSV-1 (MOI = 1) for 24 h. Quantitative RT-PCR and western blotting were performed to measure NR2F6 level. The levels of c-Jun and NR2F6 were measured by immunoblotting. (**E**) The UCSC genome browser view to show c-Jun enrichment on *Nr2f6* promoter. (**F**) c-Jun-FB stably expressed MEF cells were infected with HSV-1 (MOI = 1) for 8 h, and capture-ChIP was performed to measure c-Jun-FB enrichment on *Nr2f6* promoter. The primer target loci shown is (E). Graphs show mean ± SEM, n = 3. ** stands for p < 0.01, * for p < 0.05.

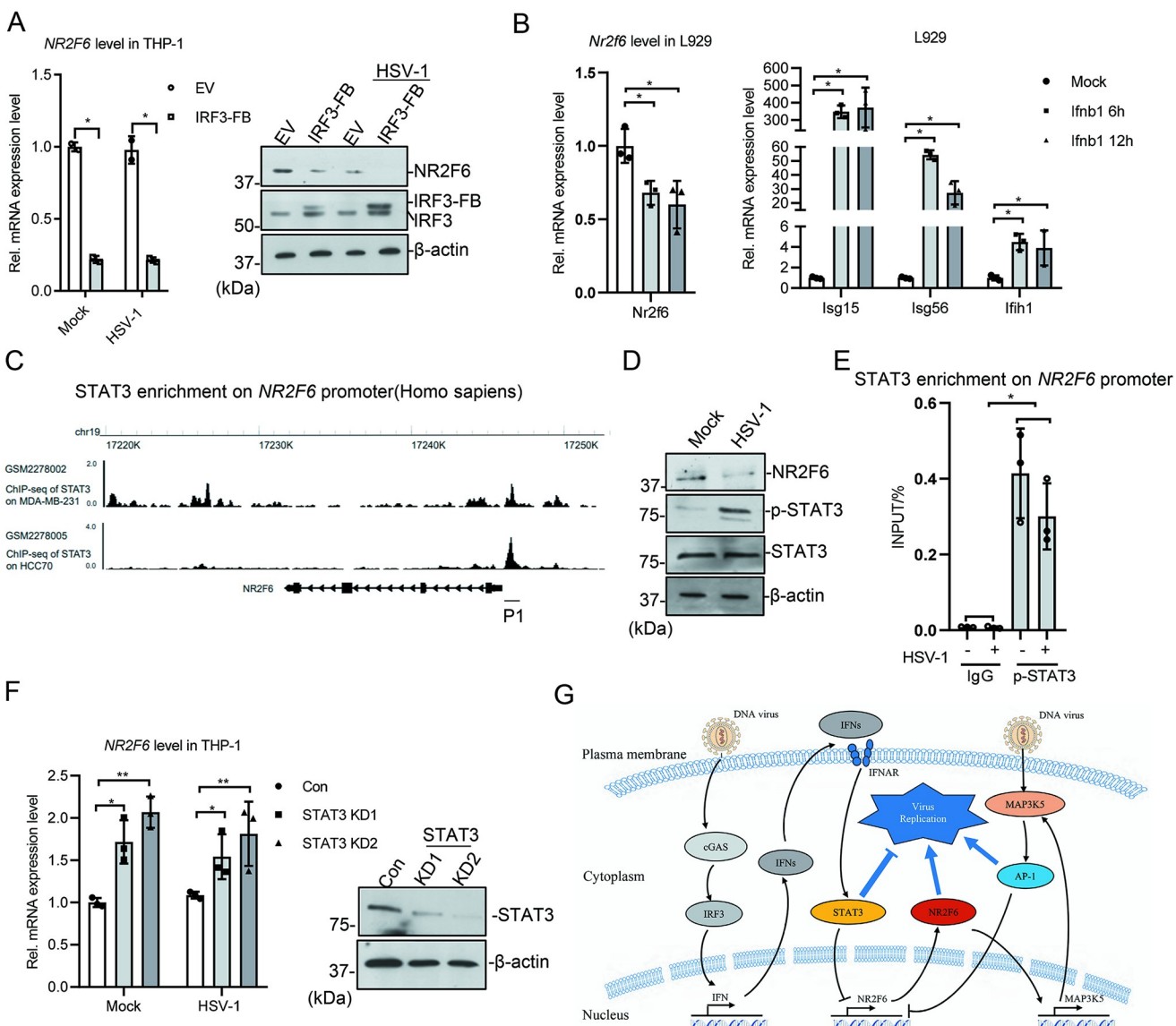

**Fig 7. STAT3 represses *NR2F6* transcription. (A)** FB-tagged *IRF3* was exogenous expressed in THP-1 cells. The cells were then infected with HSV-1 (MOI = 1) for 24 h, and NR2F6 level was measured with western blotting and qRT-PCR. **(B)** L929 cells were treated with IFN-β (100 ng/mL) for the indicated time, and the expression of *Nr2f6*, *Isg15*, *Isg56* and *Ifih1* was measured with qRT-PCR. **(C)** The UCSC genome browser view to show STAT3 enrichment on *NR2F6* promoter. **(D)** THP-1 cells were infected with HSV-1 (MOI = 1) for 24 h, and the indicated proteins were measured with immunoblotting. **(E)** THP-1 cells were infected with HSV-1 (MOI = 1) for 24 h, and ChIP-qPCR result to show the STAT3 enrichment on *NR2F6* promoter region. **(F)** STAT3 was knocked down in THP-1 with CRISPR. The cells were infected with HSV-1 (MOI = 1) for 24 h before immunoblotting and qPCR analyses. **(G)** A sketch to show the function and regulation of NR2F6 during HSV-1 infection. Graphs show mean ± SEM, n = 3. ** stands for p < 0.01, * for p < 0.05.

factors involved in viral infection and anti-viral response have been identified, such as cGAS/ STING and JAK/STAT pathways, the virus-host interaction is still not fully clear. The activation and repression of signaling pathways often lead to the change of transcription factor activity, which is also associated with the change of enhancer activity. Then the transcription factors involved in anti-viral response, can be predicted through profiling of active enhancers, which may serve as a powerful tool to identify new regulators and pathways. In the current

study, we have identified NR2F6 as a new transcription factor involved in the above process based on genome-wide enhancer profiling. We have also predicted and experimental proved MEF2D and MAFF are involved in HSV-1 infection. The studies about these new transcription factors will shed new light on anti-viral research.

NR2F6 is previously known as an orphan nuclear receptor involved in anti-tumor immunity, and genetic ablation of NR2F6 in melanoma conferred a more effective response to PD-1 therapy [35,36]. NR2F6 is also involved in the regulation of T cells by transcription regulation [37,38]. Here we show that NR2F6 is a critical factor involved in viral infection and the interaction between viruses and host cells (Fig 7G). NR2F6 promotes the replication of DNA viruses HSV-1 and MCMV, as well as RNA virus SeV. NR2F6 deficiency caused repression of virus replication, not through the classical cGAS/STING signaling, but through repression of AP-1/c-Jun pathway. NR2F6 promotes AP-1/c-Jun activation via directly targeting multiple genes involved in JNK cascade, including *MAP3K5*, a key kinase for AP-1/c-Jun pathway.

On the other hand, although viruses use AP-1/c-Jun signaling to promote their replication, host cells use AP-1/c-Jun signaling to repress *NR2F6* expression (Fig 7G). Thus, AP-1/c-Jun pathway has both anti-viral and pro-viral functions during HSV-1 infection. Several important molecules have been shown their double-edge effects during virus infection, such as G3BP1, the core protein for stress granule [42], AP-1/c-Jun pathway probably has the similar effects. Inhibition of AP-1/c-Jun pathway by two different small chemicals both repressed viral replication, indicating that upon pathway inhibition, NR2F6 lost its ability in pro-viral activity, no matter its expression was increased.

We have revealed that cGAS/STING pathway can repress NR2F6 expression through STAT3 signaling (Fig 7F). STAT3 knockdown enhanced NR2F6 expression, even in the cells without virus infection. ChIP-PCR results also showed that STAT3 binds to *NR2F6* in control cells. The IRF3/IFNs/STAT3 signaling might be another mechanism to control *NR2F6* transcription. Moreover, we found that the patterns for NR2F6 mRNA and proteins were different with HSV-1 infection. NR2F6 protein kept decreasing after infection, while its mRNA decreased after infection and then increased after a while (Fig 6B). The inconsistency suggests that post-translational modifications on NR2F6 may exist to regulate its stability. Interestingly, we found that the molecular weight of the nuclear and cytosolic NR2F6 is different (Fig 5C), suggesting post-translational modifications or alternative splicing may regulate its localization.

Taken together, we have identified three transcription factors involved in anti-viral response through an epigenomic approach. NR2F6 promotes virus replication through AP-1/c-Jun pathway, and NR2F6 expression is repressed by AP-1/c-Jun and STAT3. Our work has identified NR2F6 as an important transcription factor involved in virus infection and virus-host interaction.

## Materials and methods

### Ethics statement and animal housing

Mice were maintained in the special pathogen-free facility of Animal Center of College of Life Sciences at Wuhan University. Mice were born and maintained under pathogen-free condition at 20~24°C with a humidity of 40~70% and a 12/12-hours dark/light cycle (lights on at 7:00 am, lights off at 7:00 pm), with free access of water and food. All the animal operations followed the laboratory animal guidelines of Wuhan University and were approved by the Animal Experimentations Ethics Committee of Wuhan University (Protocol NO. 14110B). No patient study was involved and the consent to participate is not applicable.

## Reagents and cell lines

Antibodies recognizing H3 (Abcam, ab1791, RRID: AB_302613), GAPDH (Abclonal Cat# AC002, RRID:AB_2736879), β-actin (Abclonal, AC026), HA (ORIGENE cat#TA100012), NR2F6 (Abcam Cat# ab65012, RRID:AB_2155633), TBK1 (Cell Signaling Technology Cat# 3504, RRID:AB_2255663), p-TBK1 (Cell Signaling Technology Cat# 70483, RRID: AB_2943237), IRF3 (EPITOMICS cat#2241–1), p-IRF3 (S396, Cell Signaling Technology Cat#4947S), JUN (CST, 9165, RRID:AB_2130165), p-c-JUN (S63, BD Cat#558036), JNK (epitomics Cat#3496–1), p-JNK (Cell Signaling Technology Cat# 4668, RRID:AB_823588), H3K27ac (Abcam, ab4729, RRID: AB_2118291), p-STAT3(Cell Signaling Technology Cat#9145), MEF2D (Abcam Cat# ab32845, RRID:AB_776269), and MAFF (BAIJIA Cat#IPB4267) were purchased from the indicated commercial sources. Protein G-Sepharose beads (GE Healthcare) and Dynabeads MyOne streptavidin C1 (Thermo-Fisher Scientific) were purchased from the indicated companies. PCR primers were custom synthesized by Tsingke Biotechnology and siRNAs by GenePharma and JTS Scientific. SP600125 and T-5224 were purchased from Sigma. HSV-1 (KOS strain) was gifted by Dr. Hong-Bing Shu of Wuhan University, which was originally got from China Center for Type Culture Collection, Wuhan, China. HFF, THP-1, HEK293T, U2OS and L929 were purchased from the Cell Bank of Chinese Academy. HACAT was gifted from Dr. Yu Liu, and MLF was from Hong-Bing Shu of Wuhan University. U2OS and THP-1 cells were cultured in RPMI 1640 supplemented with 10% FBS, 1% penicillin, and streptomycin. All the other cells were cultured in DMEM supplemented with 10% FBS, 1% penicillin, and streptomycin. All the cells were cultured at 37°C with 5% CO2.

## Mice model and animal housing

The $Nr2f6^{+/-}$ mice were generated by CRISPR/Cas9-mediated genome editing by Beijing Biaosetu Gene Biotechnology Co., LTD. The F1 $Nr2f6^{+/-}$ mice were crossed with wild-type C57BL/ 6 mice for at least three generations. Mice were genotyped by PCR analysis and the resulted $Nr2f6^{+/-}$ mice were crossed to generate $Nr2f6^{+/+}$ and $Nr2f6^{-/-}$ mice. Age-matched and sex-matched littermates were blindingly randomized into groups for animal studies. MEF was isolated from mouse embryos in the lab. Bone marrow cells were isolated from femurs of $Nr2f6^{+/+}$ and $Nr2f6^{+/-}$ mice. M-CSF were added to the bone marrow culture for differentiation of BMDMs. Primers for Genotype identification were in S3 Table.

## ChIP assay

ChIP assay was performed as previously described [43]. Briefly, cells were cross-linked with 1% formaldehyde for 10 min at room temperature, quenched with 0.125 M glycine for 5 min. After washed twice with PBS, Cells were collected by centrifugation, and re-suspended with ChIP digestion buffer (50 mM of Tris-HCl, pH 8.0, 1 mM of CaCl2, 0.2% Triton X-100). Chromatin was sonicated to 200–500 bp at 0°C. Supernatant was equally divided after diluted with five times of dilution buffer (20 mM of Tris-HCl, pH 8.0, 150 mM of NaCl, 2 mM of EDTA, 1% Triton X-100, 0.1% SDS). Protein G beads were washed by BSA with salmon DNA twice, incubated with lysates and antibodies at 4°C overnight, and washed once with wash buffer I (20 mM Tris-HCl, pH 8.0, 150 mM NaCl, 2 mM EDTA, 1% Triton X-100, 0.1% SDS), once with wash buffer II (20 mM Tris-HCl pH 8.0, 500 mM NaCl, 2 mM EDTA, 1% Triton X-100, 0.1% SDS), once with wash buffer III (10 mM Tris-HCl pH 8.0, 250 mM LiCl, 1 mM EDTA, 1% Na-deoxycholate, 1% NP-40) and twice with TE (10 mM Tris-HCl pH 8.0, 1 mM EDTA). The beads were eluted twice with 100 μL elution buffer (1% SDS, 0.1 M NaHCO3, 0.2 mg/mL Proteinase K) at room temperature. The elution was incubated at 65 °C for 6 h and DNA was

purified (TIANGEN DP214-03). The resulted DNA was assayed by quantitative PCR. Primers were in S3 Table.

## Capture-ChIP

$2 \times 10^7$ THP-1 or MEF stable cells were harvested, cross-linked with 2 mM SMCC (MCE) for 35 min and 0.2% formaldehyde for 10 min at room temperature in PBS (pH 8.0), and then quenched with 0.125 M glycine for 5 min. Cross-linked cells were washed twice with PBS, then collected by centrifugation. Cells were lysed in 500 μL lysis buffer (0.5% SDS, 10 mM EDTA, 50 mM Tris-HCl, pH 8.0), and rotated for 20 min at 4°C. Lysates were sonicated to 200–500 bp and centrifuged at 16,100 g for 10 min at 4°C. 450 μl supernatant was transferred to a new tube and NaCl solution was added to the final concentration of 300 mM. Dynabeads MyOne streptavidin C1 (Thermo-Fisher Scientific) were washed twice with BSA and salmon extract DNA. Supernatant was incubated with 10 μL of Dynabeads at 4 °C overnight. Then the Dynabeads were washed twice with 2% SDS, twice with RIPA buffer containing 0.5 M NaCl, twice with LiCl buffer (250 mM LiCl, 0.5% NP-40, 0.5% sodium deoxycholate, 1 mM EDTA and 10 mM Tris-HCl, pH 8.0), and twice with TE buffer (10 mM Tris-HCl, 1 mM EDTA, pH 8.0). The chromatin was eluted in SDS elution buffer (1% SDS, 10 mM EDTA, 50 mM Tris-HCl, pH 8.0) followed by incubation at 65°C overnight. DNA was treated with RNase A (5 mg/mL, Sigma) and protease K (0.2 mg/mL, Biosharp) at 37°C for 30 min, and extracted with DNA purification kit (TIANGEN). The purified DNA was assayed by quantitative PCR. Primers for were listed in S3 Table.

## Library preparation for ChIP-sequencing

ChIP-seq libraries were constructed with ChIP and input DNA using VATHS Universal DNA Library Prep Kit for Illumina (Vazyme ND606). Briefly, 50 μL of DNA (8–10 ng) was end-repaired for dA tailing, followed by adaptor ligation. Each adaptor was marked with an 8-bp DNA barcode. Adaptor-ligated DNA was purified by AMPure XP beads (1:1) and then amplified by PCR of 9 cycles with the primer matching with adaptor universal part. Amplified DNA was purified again using AMPure XP beads (1:1) in 35 μL EB elution buffer. For multiplexing, libraries with different barcodes were mixed with equal molar quantities (30–50 million reads per library). Libraries were sequenced by Illumina Nova-seq platform with pair-end reads of 150 bp.

## Library preparation for RNA-sequencing

RNA-seq libraries were constructed by NEBNext Poly(A) mRNA Magnetic Isolation Module (NEB E7490) and NEBNext Ultra II Non-Directional RNA Second Strand Synthesis Module (NEB E6111). mRNA was purified with poly-T magnetic beads and first and second strand cDNA was synthesized. The resulted cDNA was purified by AMPure XP beads (1:1) and eluted in 50 μL nucleotide-free water. The subsequent procedures were the same as described in ChIP-seq library construction, except that the sequencing depth was 20 million reads per library. RNA-seq libraries were sequenced by Illumina Nova-seq platform with pair-end reads of 150 bp.

## ChIP-seq data processing

The adaptor sequence was removed using Cutadapt (version 1.16) to clean ChIP-seq raw data. Cleaned reads were mapped into human reference genome (hg19) using BWA (version 0.7.15) with default settings. Peak calling for tissues was performed by MACS2 with a p-value

threshold of 1E-8. We calculated the normalized FPKM as the ChIP-seq signal in specific region. Then HOMER (http://homer.ucsd.edu/homer/index.html,v4.9,2-20-2017) annotate-Peaks.pl was used to annotate ChIP-seq peaks compare to reference genome hg19. HOMER findMotifsGenome.pl was used to find out significant enriched motif. Replicates of ChIP-seq data were pooled for downstream analysis.

## Transcript factor enrichment

We used the findMotifsGenome.pl module in HOMER (version 4.11, http://homer.ucsd.edu/homer/, parameters were -size 200 -mask) to identify transcript factor motifs. p-value < = 0.05.

## RNA-seq data processing and DEG identification

The adaptor sequence was removed using Cutadapt (version 1.16) to clean RNA-seq raw data. Cleaned reads were aligned to the human reference genome (hg19) using HISAT2 (2.1.0) with default settings. Uniquely aligned reads were counted at gene regions using the package feature Counts (version 1.4.6) based on Gencode v19 annotations. Differential gene expression analysis was performed using the R/Bioconductor package DESeq2 (version 1.26.0) with contrast adjustment for multiple groups comparison. Genes whose log2FC < 1 and p-value < 0.05 were identified as differential expressed genes (DEGs).

## ChIP-Seq and RNA-Seq data visualization

Genomic track for histone marks, RNA expression and chromatin state beyond the RefSeq gene model were drew by UCSC browser.

## Cell line construction and lentiviral infection

The single guide RNA (sgRNA) sequences were designed by using the CRISPR Design Tool (http://tools.genomeengineering.org), provided by Feng Zhang lab. The sgRNAs were cloned in lentiCRISPRv2-puro (Addgene, #98290). To construct knockdown cell lines, the lentiviral packaging plasmids, psPAX2 and pMD2G, were transfected into HEK293T cells with plenti-CRISPR plasmid. Then the supernatant was used to infect the desired cells, and cells were selected by puromycin. Infection with THP-1 requires centrifugation at 1000 g for 30 min. The sgRNA sequences were listed in S3 Table.

## Western blotting

Cells were lysed in SDS loading buffer containing 50 mM Tris pH 6.82, 4% SDS, 0.2% bromo-phenol blue, 10% glycerol, and 5% β-mercaptoethanol. The lysates were heated at 95˚C for 5 minutes, and then subjected to electrophoresis using SDS-PAGE gels. After transferring to nitrocellulose membrane, 5% milk in TBST buffer was used to block the membrane at room temperature for 1 h. Primary antibodies and of HRP-conjugated secondary antibodies were incubated with the blot for 1 h in 5% milk/TBST. The membrane was washed and photographed with ECL.

## Reverse transcription and quantitative PCR

Total RNA was extracted from cells or tissues using TRIzol (Aidlab. biotech), and the first-strand cDNA was reverse-transcribed with All-in-One cDNA Synthesis SuperMix (Vayzme). Total DNA was extracted from cells or tissues using TRIzol (TIANamp). Gene expression was examined with a BioRad CFX Connect system by a fast two-step amplification program with

2 × SYBR Green Fast qRT-PCR Master Mix (Vayzme). The value obtained for each gene was normalized to that of the gene encoding β-actin. The sequences of primers used in this study were listed in S3 Table.

## Viral infection

For qRT-PCR analysis, cells were seeded into 6-well plates (2–4 × 10$^5$ cells per well) and infected with the specified viruses for the indicated time points. For viral replication assays, cells (2–4 × 10$^5$) were infected with HSV-1 or SeV followed by culture in full medium for 12 h or 24 h. Viral transcription and replication was analyzed by qPCR analysis. For the infection of mice, 8-12-week-old male were intraperitoneal injected with HSV-1 (5 x 10$^7$ PFU per mouse). The survival of animals was monitored every day. The liver and lung tissues (n = 5) were collected for RT-qPCR at 48 h post-injection.

## Plaque assay

The supernatants of THP-1 or MEFs cultures were used to infect monolayers of Vero cells. One hour later, the supernatants or homogenates were removed and the infected Vero cells were washed with pre-warmed PBS twice followed by incubation with DMEM containing 2% methylcellulose for 48 h. The cells were fixed with 4% paraformaldehyde for 15 min and stained with 1% crystal violet for 30 min before counting the plaques.

## Immunofluorescent staining

Cells were cultured on the cover slips and fixed with freezing methanol after washed twice in PBS. The cover slips were then washed three times by PBS and blocked in PBS with 1% BSA and 0.1% NP-40 for 10 min. The cover slips were hybridized with first and second antibodies for 1 h, respectively. Then the slips were mounted with prolong anti-fade kit (Invitrogen) and observed with confocal fluorescent microscopy.

## Cell fractionation

The cells were digested into EP tubes with trypsin and washed twice with PBS. Cells were resuspended with 500 μl buffer A (5 mM MgCl2, 10 mM NaCl, 1 mM DTT, 100 mM Tris-HCl, pH 7.4) and incubated on ice for 15–20 min. NP40 was added to the final concentration of 0.5% followed by oscillate at high speed. Then the mix incubated on ice for 1min followed by oscillate at high speed. The mix was centrifuged at 500 g for 5 min at 4˚C. The supernatant was centrifuged at 16000 g for 30 min at 4˚C, and the resulted supernatant is cytoplasmic fraction. The precipitate was suspended with 250 μl buffer A, incubated on ice for 5–10 min and then centrifuged 500 g for 5 min at 4˚C. The resulted precipitate is nuclear fraction.

## Statistical analysis

For all the experimental studies, the assays were repeated at least three times. At least two biological replicates were used for NGS studies; and for animal studies, at least five biological replicates were used. Data were shown as average values ± SD and p value. Student's t-test was used for comparison between groups.

## Supporting information

**S1 Fig. Other anti-virus transcription factor candidates affected HSV-1 transcription.** Effects of deficient Mef2d (A) and Maff (B) on HSV-1 transcription in MEF cells. The MEF cells were treated with siRNA for 48 h before infection with HSV-1. The MEF cells were

infected with or without HSV-1 (MOI = 1) for 8h before qPCR analysis. (C) Effects of deficient NR2F6 on HSV-1 in THP-1. The THP-1 cells were infected with different virus titers for 24h (MOI = 0.25, 0.5, 1). (D) Effects of deficient NR2F6 on HSV-1 in THP-1. The THP-1 cells were infected with HSV-1(MOI = 1) for different times (8h, 24h). Graphs show mean ± SEM, n = 3. **P < 0.01, *P < 0.05.
(PDF)

**S2 Fig. NR2F6 can affect the transcription and replication of multiple viruses in a variety of cells.** Effects of deficient NR2F6 on HSV-1 and SeV transcription in U2OS(A) and HFF(B). The cells were transfected with siRNA for 48h before infection with HSV-1 or SeV. The cells were left uninfected or infected with HSV-1 (MOI = 1) for 8h or SeV for 12h before qPCR analysis. Effects of overexpressed NR2F6 on SeV transcription in MEF(C) and HEK293T(D). The cells were transfected with plasmids for 48h before infection with SeV. The cells were left uninfected or infected with SeV for 12h before analysis. (E) Effects of deficient Nr2f6 on MCMV transcription and replication in MEF. The MEF cells were left uninfected or infected with MCMV (MOI = 1) for 72h before qPCR analysis. Graphs show mean ± SEM, n = 3. **P < 0.01, *P < 0.05.
(PDF)

**S3 Fig. The impact of Nr2f6 deficiency on mouse development.** (A)The Nr2f6 gene in mice were mutated using CRISPR system. (B) The genome identification results in different genotypes. (C) Transcription level of Nr2f6 in different mouse genotypes. The weight of spleen(D), lung (E) and liver (F) of wide type and Nr2f6+/- mice was measured.
(PDF)

**S4 Fig. Effects of NR2F6 on gene transcription of anti-virus innate immunity.** (A) Effects of NR2F6 deficiency on transcription of downstream genes(FOS, IL8, ISG56) induced by HSV-1 in THP1 cells. The THP1 cells were infected with HSV-1 (MOI = 1) for 24 h before qPCR analysis. (B) Effects of overexpressed Nr2f6 on TBK1-IFNB1 and JNK passway in MEF cells. The cells were transfected with plasmids for 48 h before infection with HSV-1. The cells were left uninfected or infected with HSV-1 (MOI = 1) for 8 h before immunoblotting analysis. (C) Effects of overexpressed NR2F6 on transcription of downstream genes(IFNB1, ISG54, ISG56) induced by HSV-1 in THP1 cells. The THP1 cells were infected with HSV-1 (MOI = 1) for 24 h before qPCR analysis. Graphs show mean ± SEM, n = 3. **P < 0.01, *P < 0.05.
(PDF)

**S5 Fig. AP-1 affected gene transcription of anti-virus innate immunity.** (A) Effects of deficient c-Jun on anti-virus innate immunity genes transcription with HSV-1 infection. The THP1 cells were infected with HSV-1 (MOI = 1) for 24 h before qPCR analysis. (B) Effects of deficient c-Jun on SeV replication and anti-virus innate immunity genes (IFNB1, ISG54, ISG56) transcription with SeV infection. The THP1 cells were infected with SeV for 12 h before qPCR analysis. Graphs show mean ± SEM, n = 3. **P < 0.01, *P < 0.05.
(PDF)

**S6 Fig. NR2F6 directly regulates the transcription of UBC and TRIB1.** (A) NR2F6 affected the transcription of UBC in THP-1 cells. The THP1 cells were infected with HSV-1 (MOI = 1) for 24 h before qPCR analysis. (B) The UCSC genome browser view showed that NR2F6 was enriched on the UBC promoter region. (C) The enrichment of NR2F6 on UBC promoter region was detected in NR2F6-FB overexpressed THP-1 cells by ChIP-qPCR. The THP-1 cells were infected with HSV-1 (MOI = 1) for 24 h before analysis. (D) NR2F6 affected the transcription of TRIB1 in THP-1 cells. The THP1 cells were infected with HSV-1 (MOI = 1) for 24

h before qPCR analysis. (E) The UCSC genome browser view showed that NR2F6 was enriched on the TRIB1 promoter region. (F) The enrichment of NR2F6 on TRIB1 promoter region was detected in NR2F6-FB overexpressed THP-1 cells by ChIP-qPCR. The THP-1 cells were infected with HSV-1 (MOI = 1) for 24 h before analysis. Graphs show mean ± SEM, n = 3. $^{**}$P < 0.01, $^{*}$P < 0.05.
(PDF)

**S1 Table. The expression of all the genes in control and HSV-1-treated THP-1 cells.**
(XLSX)

**S2 Table. The expression of all genes in control and NR2F6 knockdown cells.**
(XLSX)

**S3 Table. Sequence information of sgRNA, siRNA and primers.**
(XLSX)

## Acknowledgments

We appreciate Dr. Hong-Bing Shu of Wuhan University for sharing reagents and project discussion.

## Author Contributions

**Conceptualization:** Min Wu, Lian-Yun Li.

**Formal analysis:** Chen-Yu Wang, Qiao-Yun Long, Ming-Liang Wei.

**Funding acquisition:** Ming-Kai Chen, Min Wu, Lian-Yun Li.

**Investigation:** Chen Yang.

**Methodology:** Shan-Bo Tang, Xiang Lin.

**Project administration:** Yong Xiao, Ming-Kai Chen, Min Wu, Lian-Yun Li.

**Resources:** Zhuo Cao, Xiang Lin, Zi-Qi Mu.

**Supervision:** Min Wu, Lian-Yun Li.

**Visualization:** Chen-Yu Wang, Qiao-Yun Long, Ming-Liang Wei.

**Writing – original draft:** Chen Yang, Min Wu.

**Writing – review & editing:** Chen Yang, Chen-Yu Wang, Yong Xiao, Ming-Kai Chen, Min Wu, Lian-Yun Li.

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
