## [Decision Letter · Decision Letter 0]

12 Mar 2024

Dear Dr. Wu

Thank you very much for submitting your manuscript "The roles of nuclear orphan receptor NR2F6 in anti-viral innate immunity" for consideration at PLOS Pathogens. As with all papers reviewed by the journal, your manuscript was reviewed by members of the editorial board and by several independent reviewers. In light of the reviews (below this email), we would like to invite the resubmission of a significantly-revised version that takes into account the reviewers' comments.  As you can see from the attached reviews, all three reviewers had a significant number of comments/questions and suggestions that would make the manuscript a clearer and stronger manuscript.  It is important that you experimentally address these comments from the reviewers and provide a point by point response to reviewers outlining all of the changes that were made in the revised manuscript.  In addition, reviewer 1 raised concerns regarding data inconsistency in some of the figures.  It is critical that these discrepancies be resolved and the appropriate Figures be included in the revision.  

We cannot make any decision about publication until we have seen the revised manuscript and your response to the reviewers' comments. Your revised manuscript is also likely to be sent to reviewers for further evaluation.

Sincerely,

Donna M Neumann

Academic Editor

PLOS Pathogens

Alison McBride

Section Editor

PLOS Pathogens

Michael Malim

Editor-in-Chief

PLOS Pathogens

orcid.org/0000-0002-7699-2064

Reviewer's Responses to Questions

**Part I - Summary**

Reviewer #1: In this study, Yang et al. reports NR2F6 as a new DNA viral infection transcriptional regulator in promoting DNA viral replication. Using RNA-Seq and H3K27ac ChIP-Seq in THP1 cells upon HSV-1 infection, the authors found enriched TF motifs upon viral infection. By validating hits, the authors found NR2F6 facilitated DNA viral replication. By generating Nr2f6 hetero-mice, the authors confirmed a positive regulator role of Nr2f6 in facilitating viral replication in vivo. Further mechanistic studies reveal involvement of JNK/c-Jun with Nr2f6 in regulating MAP3K5 transcription, while a negative suppression loop by IRF3/STAT3 has also been observed. Overall, the authors provide ample amount of data to indicate the presence of a new NR2F6 transcriptional program in facilitating DNA viral infection. However, there are severe technical concerns and some conceptual concerns that preclude the study to be considered for publication.

Technical concerns: Inconsistent data have been presented across the data panels that reduce the confidence of the reviewer on this study. eg. NR2F6 mRNA levels are not consistent for with/without HSV-1 infection in Fig. 1F (decreased upon HSV-1) vs Fig. 7F (no significance); the NR2F6 western blots in Fig. 6B (doublets) vs. Fig. 6D and many other panels (single band).

Conceptual concerns: If NR2F6 is facilitating DNA viral replication and infection, its suppression by STAT3 should occur in later infection stage- why at 24 hr infection increased NR2F6 genome localization is observed (by H3K27Ac) but the overall NR2F6 expression is reduced? And time-dependent NR2F6 has not been established.

Reviewer #2: I think this work is generally robust, interesting, and important. My biggest qualms about this manuscript have to do with the presentation of the story and data, as well as physiological relevance of some of the cell lines used. Especially knowing how far animal models and neuronal cell culture models have come for HSV work, I found the models to be a little underwhelming.

Strengths – multiple orthogonal approaches identified pro-viral transcription factors not previously appreciated. Mapped mechanism to signaling pathway, and changes in genome occupancy of transcription factors that underlie the phenotypes presented.

Weaknesses – despite the amount of compelling data, it was difficult to understand due to the way the data were presented (see minor issues below).

Reviewer #3: In this manuscript, Yang et. al unravel a complex web of regulation centered around a transcription factor, NR2F6, that they have uncovered as part of the response to viral infection using HSV-1 as a model. First, they perform RNA-seq to identify upregulated differentially expressed genes during HSV-1 infection, then performed H3K27ac-ChIP-seq to identify active enhancers within this group. From this dataset, potential transcription factors were identified and several (Nr2f6, Mef2d and Maff) were characterized. This paper focuses on NR2F6, an orphan nuclear receptor. The authors show that knockdown of NR2F6 reduces HSV-1 replication and that overexpression enhances replication. The authors then generate Nr2f6+/- mice and show that partial loss of NR2F6 reduces viral replication in isolated BMDMs and mice. RNA-seq of NR2F6-null cells revealed that differentially expressed genes were enriched in the JNK pathway.

The authors going on to show that NR2F6 promotes HSV-1 replication not through the cGAS/STING pathway, but instead through the JNK pathway. NR2F6 works with c-Jun to bind to the MAP3K5 promoter, activating the JNK pathway. NR2F6 is in turn downregulated by both STAT3 (cGAS/STING) and the AP-1/c-Jun pathway.

The work is novel, unveiling a new piece of the innate immune system that responds to viral infection. Although HSV-1 was used in this paper, similar trends hold for MCMV and Sendai virus, suggesting that this pathway may be broadly involved in viral infection. The work is thorough and shows similar trends in vitro in many different cell types and in vivo.

**Part II – Major Issues: Key Experiments Required for Acceptance**

Reviewer #1: Specific comments:

1. The knockdown efficiency of Med2d in Fig. S1A is quite poor thus conclusions from Med2d are not conclusive.

2. Fig 1F: since Nr2f6 is the major target for this study, protein level changes would also need to be monitored in this assay in addition to mRNA changes. In addition, the statistical analysis in Nr2f6 levels between blank and HSV-1 infection groups in con should be performed- does this suggest HSV-1 infection at 24 hr time point suppresses Nr2f6 transcription? If so, how can Nr2f6 mediated enhancer-mediated transcription be activated and enriched?- also the authors should comment on why in Fig. S1A Med2d mRNA levels increased upon HSV-1 infection, which is opposite to Fig. 1F.

3. Fig. 1G: these blots were not aligned well.

4. One major concern for NRF2D6 knockdown is if this affects cell proliferation- if so proliferation changes would expect to affect viral infection. Thus, at least cell viability or colony formation assays should be performed in mouse and human lines used here.

5. Fig S3: how about the body weights for normal and hetero- mice?

6. Fig 2B: NR2F6 signals are too weak to find endogenous signals.

7. Fig 2C-2F: one of the major HSV-1 infection organ is brain, thus brain viral load should be monitored. What is the rationale to separate Fig 2D and 2F?

8. Fig 3B: it is not clear if these critical data are from biological triplicates or no duplicates? RNA-Seq on single sample does not support any conclusions.

9. Fig 3C: It seems top hits are related to cell migration- how these processes are related to anti-viral responses and NR2F6 anti-viral function?

10. Fig 3E: to demonstrate cGAS/STING pathway is not involved, cGAS or STING knockdown should be performed in THP1 cells.

11. Fig 4A: the knockdown efficiency of NR2F6 is quite low here.

12. Fig 5B: low quality IF images didn’t add much to this study. Why not detecting endogenous NR2F6? If HSV-1 infection does not affect NR2F6 cellular localization, how does HSV-1 infection regulate NR2F6 function at enhancers?

13. Fig. 5: a critical piece of information is missing here: does NR2F6 binds c-Jun or co-localizes with c-Jun by ChIP on super-enhancers defined in Fig 1? – NR2F6 ChIP-Seq would be necessary.

14. Fig 6B: why NR2F6 blot here becomes a doublet with a different pattern than other NR2F6 blots (eg. Fig 3 and Fig 4)?

15. Fig 7F: why NR2F6 mRNA levels here in THP-1 cells upon HSV-1 infection remain the same (which is opposite to Fig. 1F (decreased) ?

Reviewer #2: No major issues. The authors present a lot of data using complementary approaches that ultimately support the conclusions of the paper.

Reviewer #3: (No Response)

**Part III – Minor Issues: Editorial and Data Presentation Modifications**

Reviewer #1: N/A

Reviewer #2: While I do not think any key experiments are missing, I think that the manuscript as it stands is unclear and difficult to get through due to presentation. Here are some comments and suggestions to help guide the authors to writing a clearer manuscript:

• Why THP1 cells to start? This feels not particularly physiologically relevant considering the main latent reservoirs of HSV1 is neurons, and the general paradigm of disease involves initial infection at mucosal sites followed by neuroinvasion for establishment of latency. The manuscript would be greatly improved by either more clearly justifying the use of THP1s or performing more experiments in more relevant model cell types.

• In figure 1F, you show that late HSV1 gene expression is downregulated in the context of Nr2f6 KD, but it also looks like HSV1 infection leads to a decrease in Nr2f6 mRNA. MUCH later, in figure 6, you finally address this question in multiple cell types. I found myself thinking early on that it would be informative to know the dynamics of this protein vs RNA, especially since you’re saying it’s a transcription factor, thinking about the timing of expression to action. It might be useful to include the data from Fig6A-C early on, as early as figure 1 or 2.

• It would be extremely helpful if you reported more clearly the fold changes in RNAseq for the hits you’re following up on, especially Nrf26. Readers don’t want to have to dig through your whole dataset to see what this is. Especially because the directionality of change is not clearly noted for Nrf26, Figure 1E has a heat map with numbers but those numbers are not labeled so it is unclear what they mean.

• It would also be helpful to show where you found Nr2F6 in your ChIP seq early on – I found myself wondering where it was bound early on, and clearly you had seq data and then you validated it several figures later. I think it would be useful to show the tracks in Fig1 before following up on it rather than waiting till fig 5.

• The focus in the text on “DNA viruses” when you are clearly only showing data for HSV1 is misleading and inaccurate. Especially when later you test an RNA virus – Sendai virus. This is quite cool! But it is not in line with the story you’re telling, and to suggest that what you’ve found for HSV1 is relevant for all DNA viruses, a viral lineage with immense diversity in disease, cell tropism, genome size and composition, and mechanisms of altering host cells to promote their infectious cycles, is not appropriate. Stay specific, or test another virus. It’s fine, interesting, and fun to speculate – would be great in the discussion! But you show no evidence for it here, yet, so framing your story this way in the introduction felt disingenuous.

• You don’t mention the results for Fig 3 F ISG54 in the text at all – highlighting the difference between this and the other genes tested in this figure would help clarify your point

• It would be extremely helpful to give a little more information about what Nr2f6 is earlier in the manuscript – the discussion has info that would be extremely useful to understand the rationale of experiments earlier on.

• The leftmost panel in Fig 1F is labeled “US11 level in MEF” but the y-axis is labeled “Relative virus DNA amount” – I’m pretty sure this is qPCR for viral genomes, but would be important to make sure these figures are labeled accurately. As a general note – I would recommend going through each figure to double check all of the labels for this type of mistake, and there are several figures that don’t have keys/labels in either the figure itself or the figure legend (Fig 1E, Fig 3C)

• Fig 5F and 5H could use more informative labels than “ENCSR707QWA_1” etc – label them so readers can easily tell what they are looking at; same for Fig 6E, 7C.

• What was the rationale for following up on Nrf26 and not Mef2d or Maff? Those data buried in the supplement are also compelling – especially that HSV1 seems to have the opposite effect on Maff (upregulated RNA levels) than it does on Nrf26. Perhaps this is will be a followup study? More clarity on the rationale of why Nrf26 was the focus would be helpful for following along in the story.

• Sup Fig S1 “candidaters” typo – “candidates”

• I think if there’s a way to include it, I would have liked to see Sup Fig S4 in the main text – at least panels A-B. I understand space constraints, but this seems more important and interesting than relegating to supplement.

• While the SeV work is interesting – what does it actually add to the story here? If you want to make the point that this could be a pan-viral effect of Nrf26, it would be more impactful to spend the whole start off the paper on HSV, and then at the end, have one figure with all of the Sev data that says look it is actually seemingly impo

---

## [Editor Report · Decision Letter 1]

17 May 2024

Dear Dr. Wu

We are pleased to inform you that your manuscript 'The roles of nuclear orphan receptor NR2F6 in anti-viral innate immunity' has been provisionally accepted for publication in PLOS Pathogens.

Best regards,

Donna M Neumann

Academic Editor

PLOS Pathogens

Alison McBride

Section Editor

PLOS Pathogens

Michael Malim

Editor-in-Chief

PLOS Pathogens

orcid.org/0000-0002-7699-2064
---

## [Editor Report · Acceptance letter]

29 May 2024

Dear Dr. Wu,

We are delighted to inform you that your manuscript, "The roles of nuclear orphan receptor NR2F6 in anti-viral innate immunity," has been formally accepted for publication in PLOS Pathogens.

Best regards,

Michael Malim

Editor-in-Chief

PLOS Pathogens

orcid.org/0000-0002-7699-2064